# Multiparent-Derived, Marker-Assisted Introgression Lines of the Elite Indian Rice Cultivar, ‘Krishna Hamsa’ Show Resistance against Bacterial Blight and Blast and Tolerance to Drought

**DOI:** 10.3390/plants11050622

**Published:** 2022-02-25

**Authors:** Jyothi Badri, Gandhudi Lakshmidevi, L. R. K. JaiVidhya, Madamsetty Srinivasa Prasad, Gouri Shankar Laha, Vattikutti Jhansi Lakshmi, Subhakara Rao Isetty, Revadi Padmashree, Divya Balakrishnan, Yasaswini Vishnu Priya Varanasi, Aravind Kumar Jukanti, Uma Maheshwar Singh, Vikas Kumar Singh, Arvind Kumar, T. Ram, Lella Venkata Subba Rao, Raman Meenakshi Sundaram

**Affiliations:** 1ICAR-Indian Institute of Rice Research (ICAR-IIRR), Rajendranagar, Hyderabad 500030, India; lakshmi.agbiotech@gmail.com (G.L.); vidhyakrishnan3412@gmail.com (L.R.K.J.); data.msprasad@gmail.com (M.S.P.); lahags66@gmail.com (G.S.L.); jhansidrr@gmail.com (V.J.L.); subhakarrao.i@gmail.com (S.R.I.); padmashreerevadi@gmail.com (R.P.); dbiirr23@gmail.com (D.B.); yasaswinivishnupriya@gmail.com (Y.V.P.V.); aravindjukanti@gmail.com (A.K.J.); t.ram2011@yahoo.com (T.R.); lvsubbarao1990@gmail.com (L.V.S.R.); rms_28@rediffmail.com (R.M.S.); 2International Rice Research Institute South Asia Regional Centre (ISARC), Varanasi 221106, India; uma.singh@irri.org (U.M.S.); arvind.kumar@cgiar.org (A.K.); 3International Rice Research Institute (IRRI) South-Asia Hub, ICRISAT Campus, Hyderabad 502324, India; v.k.singh@irri.org; 4International Crop Research Institute for Semi Arid Tropics (ICRISAT), Hyderabad 502324, India

**Keywords:** rice, Krishna Hamsa, marker-assisted introgression, bacterial leaf blight, blast, drought, intercrossing, forward breeding

## Abstract

Major biotic stresses viz., bacterial blight (BB) and blast and brown plant hopper (BPH) coupled with abiotic stresses like drought stress, significantly affect rice yields. To address this, marker-assisted intercross (IC) breeding involving multiple donors was used to combine three BB resistance genes—*xa5, xa13* and *Xa21*, two blast resistance genes—*Pi9* and *Pi54*, two BPH resistance genes—*Bph20* and *Bph21*, and four drought tolerant quantitative trait loci (QTL)—*qDTY1.1, qDTY2.1, qDTY3.1* and *qDTY12.1*—in the genetic background of the elite Indian rice cultivar ‘Krishna Hamsa’. Three cycles of selective intercrossing followed by selfing coupled with foreground selection and phenotyping for the target traits resulted in the development of 196 introgression lines (ILs) with a myriad of gene/QTL combinations. Based on the phenotypic reaction, the ILs were classified into seven phenotypic classes of resistance/tolerance to the following: (1) BB, blast and drought—5 ILs; (2) BB and blast—10 ILs; (3) BB and drought—9 ILs; (4) blast and drought—42 ILs; (5) BB—3 ILs; (6) blast—84 ILs; and (7) drought—43 ILs; none of the ILs were resistant to BPH. Positive phenotypic response (resistance) was observed to both BB and blast in 2 ILs, BB in 9 ILs and blast in 64 ILs despite the absence of corresponding R genes. Inheritance of resistance to BB and/or blast in such ILs could be due to the unknown genes from other parents used in the breeding scheme. Negative phenotypic response (susceptibility) was observed in 67 ILs possessing BB-R genes, 9 ILs with blast-R genes and 9 ILs harboring QTLs for drought tolerance. Complex genic interactions and recombination events due to the involvement of multiple donors explain susceptibility in some of the marker positive ILs. The present investigation successfully demonstrates the possibility of rapid development of multiple stress-tolerant/resistant ILs in the elite cultivar background involving multiple donors through selective intercrossing and stringent phenotyping. The 196 ILs in seven phenotypic classes with myriad of gene/QTL combinations will serve as a useful genetic resource in combining multiple biotic and abiotic stress resistance in future breeding programs.

## 1. Introduction

Rice, an important crop for human sustenance, occupies a prominent place in the Indian agriculture. It is the cheapest source of calories for the developing countries and is a staple food crop for more than half of the global population, despite the changing climatic, social and economic scenario. The major biotic stresses like bacterial leaf blight (BB), blast, sheath blight, brown plant hoppers (BPH), stem borer, etc., result in a severe yield reduction in rice. In addition, abiotic stresses *viz*., drought, salinity, temperature extremities, etc., hinder growth and development of rice plant [1]. Development of climate resilient varieties with multiple stress tolerance is needed for preventing yield losses and increasing the income of rice farmers in an environmentally sustainable manner.

Conventional and marker-assisted backcross breeding has traditionally been used to introduce useful agronomic traits into elite cultivars; however, combining high yield with multiple stress tolerance using these approaches is tedious. The simultaneous occurrences of multiple abiotic and biotic stresses have demanded the development of climate-smart rice by combining quantitative trait loci (QTL) and genes for tolerance or resistance to various stresses in the genetic background of high yielding cultivars to confer a wider range of tolerance or resistance [1]. Enhanced capability of climate-smart cultivars would enable the crop to thrive under adverse environmental conditions. During the last 40 years, molecular marker systems have become well established, enabling precision in selection. Gene/QTLs conferring resistance/tolerance for various biotic/abiotic stresses are well characterized, and the recent advances in molecular marker technology and genomics have played an important role in developing single [2,3] and multiple stress-tolerant rice cultivars [1,4,5,6].

Among the biotic stress, BB caused by *Xanthomonas oryzae* pv. *oryzae* with about 22 pathotypes identified from diverse geographies [7] significantly reduces the rice yields. To date, at least 45 BB resistance genes [8,9] have been identified, and 11 of them have been cloned (*Xa1, xa5, xa10, xa13, Xa21, Xa23, Xa25* and *Xa27* [10,11]) and characterized [12], and 7 genes (*Xa4, Xa7, Xa22, Xa30, Xa31, Xa33* and *Xa34*) have been fine-mapped. Some of them have been introgressed in genetic background of elite cultivars and few of these have been released as cultivars. The obvious difference in the level of resistance among the genes to a number of virulent pathogens encouraged plant breeders to pyramid two or more genes. *Xa4* from TKM6 and *xa5* from DZ192 [13], *xa13* from long grain [14] and *Xa21* from *O. longistaminata* [15] were used as donors in the development of near isogenic lines (NILs)-IRBB60 (*Xa4*+*xa5+xa13+Xa21*) and IRBB62 (*xa5+xa13+Xa21*), with four and three genes introgressed, respectively, in the genetic background IR24 [16]. Further, an improved version of ‘PR106’ an elite cultivar with *xa5+xa13+Xa21* introgression was developed from IRBB62 [17] and improved PR106 (SS1113) was used in the transfer of *xa5+xa13+Xa21* into the background of Samba Mahsuri, a fine grain medium slender grain type variety quite popular in Southern India, which resulted in the development of NILs ‘RP Bio226’ released as a variety in the name of ‘Improved Samba Mahsuri’ (ISM) with *xa5+xa13+Xa21* [1]. The incorporation of *Xa33* alone in the background of ‘Akshayadhan’ and *Xa38*, in combination with *xa5, xa13* and *Xa21* in the background of Improved Samba Mahsuri, has proved to provide broad-spectrum resistance to BB, and the same varieties have been released as DRR Dhan 58 [18] and DRR Dhan 53 [7,19], respectively.

Blast, caused by the fungus *Magnaporthe oryzae* Barr, is the most devastating fungal disease in rice causing up to 50% yield losses [20]. It is also referred as rice fever disease and has been reported in approximately 85 rice growing countries across the world [21]. Several (>100) blast-R genes have been identified and 31 genes (*Pi37, Pit, Pish, Pi35, Pi64, Pib, pi21, Pi63/Pikahei-1(t), Pid2, Pi9, Pi2, Pizt, Pid3, Pi25, Pi50, Pigm, Pid3-I1, Pi36, Pi5, Pikm, Pb1, Pi54, Pia, Pikp, Pik, Pi1, Pi-Co39, Pike, Pita* and *Ptr*) have been cloned and characterized [22,23,24,25]. *Pi1* from West African cv. LAC23 and *Pi2* from cv. 5173 [26], *Pi9* from *O. minuta* [27] and *Pi54* from Tetep [28], when deployed in elite varietal backgrounds either singly or in combination, are effective against a wide range of blast pathotypes. Enhanced resistance to blast disease was observed in IET 25484 (RP 5960 Patho 7-5-9) with *Pi2* and IET 25480 (Pusa 1850-27) with three genes—*Pi54, Pi1* and *Pi^ta^*, now released as DRR Dhan 51 and Pusa Samba, respectively, in India [29]. Many of the resistant genes have been incorporated in modern rice cultivars [30]. These studies indicate that the use of multiple blast resistance gene combinations could be effective against blast disease.

Among the insect pests, BPH is the most devastating insect pest of rice with symptoms popularly known as hopper burn. *Bph20* and *Bph21* from *O. minuta* in the introgression line, IR 71033-121-15-B, showed BPH resistance [31]. The pyramided or single-gene introgression lines with six dominant BPH resistance genes (*Bph3, Bph14, Bph15, Bph18, Bph20* and *Bph21*) showed enhanced resistance to the recurrent parent line Jin 23B [32]. 

Water scarcity consequent to climate change is the most serious climatic challenge, particularly to the rainfed area of rice cultivation in Southern and southeastern Asia, affecting >23 million ha of rice area [33]. The reproductive stage is the most sensitive to drought stress with significant yield loss [34]. As many as 14 QTLs for drought tolerance (*qDTY1.1, qDTY2.1, qDTY3.1, qDTY6.1, qDTY3.2, qDTY12.1, qDTY2.2, qDTY4.1, qDTY9.1* and *qDTY10.1*) have been identified [35,36,37,38,39]. *qDTY1.1, qDTY2.1, qDTY3.1* and *qDTY12.1* have been effectively used in the development of rice cultivars with 10–30 yield advantage over the recurrent parents under drought stress [40,41,42]. 

Recently, efforts have been directed in sequential introgression of two or more biotic and abiotic traits. In the background of Improved Samba Mahsuri with inherent *xa5, xa13* and *Xa21*, and DRR Dhan 58 (IET 28784), DRR Dhan 60 (IET 28061) and DRR Dhan 62 (IET 28804) have been developed and released as cultivars with introgression of *Saltol* QTL for salinity tolerance, *Pup1* QTL for low soil P tolerance and *Pi2* and *Pi54* for blast resistance, respectively [6,18]. Additionally, there are successful examples of simultaneous introgression of multiple QTL and genes for biotic and abiotic stress in rice like drought and submergence tolerance in the background of Swarna [41], blast, BB, gall midge and drought tolerance in the background of Swarna [1] and Naveen [4], and blast, BB and drought tolerance in Lalat [5]. In India, *boro* season is a highly productive rice growing ecology in the Eastern and northeastern regions of the country from November to April. “Boro” means a special type of rice cultivation on residual or stored water in low-lying areas after the harvest of wet season (*kharif*) rice. Farmers have a limited choice in this ecology as only 35 cultivars have been released for cultivation to date, compared to several hundreds of cultivars in other rice growing ecologies of the country. Hence, the present study is aimed at improvement of ‘Krishna Hamsa’, an elite cultivar suitable for boro areas, by combining resistance against BB, blast, BPH and drought tolerance. Repeated cycles of intercrossing to combine traits from different donors, genotyping with foreground markers to track alleles and stringent phenotyping were deployed at different generations.

## 2. Results

### 2.1. Selective InterCrossing to Combine Genes and QTLs

In the genetic background of the elite rice cultivar, ‘Krishna Hamsa’ genes and QTLs conferring tolerance to various biotic and abiotic stresses were combined using selective intercrossing approach and forward breeding aided by marker-assisted selection. In *kharif* 2013, six simultaneous single cross F_1_’s were generated with 63 to 126 seeds by crossing ‘Krishna Hamsa’ with each of the six donors. Hybridity of 595 F_1_ plants was confirmed using gene-specific or tightly linked polymorphic SSRs for target genes and polymorphic markers at peak and flanking regions of the QTL (Appendix A). The first cycle of intercrosses were attempted during *rabi* 2014 among three pairs of single cross F_1_’s: (1) Krishna Hamsa/IRBB60 (*xa5+xa13+Xa21)*//Krishna Hamsa/IR74371-46-1-1-13 (*qDTY12.1*), (2) Krishna Hamsa/Tetep (*Pi9+Pi54*)//Krishna Hamsa/IR96321-1447-561-B-1 (*qDTY1.1+qDTY3.1*) and (3) Krishna Hamsa/IR71033-121-15-B (*Bph20+Bph21*)//Krishna Hamsa/IR74371-46-1-1-13 (*qDTY2.1*) (Figure 1).

Foreground selection in 635 IC_1_F_1_ plants revealed one to four gene/QTL in 37 different combinations of which three gene/QTL combinations*—xa5+xa13+Xa21+qDTY12.1*, *Pi9+Pi54+qDTY1.1+qDTY3.1* and *Bph20+Bph21+qDTY2.1*—were selected for crossing. In *kharif* 2014, second cycle of intercross was attempted between IC_1_F_1_ plants with gene/QTL combinations of *xa5+xa13+Xa21+qDTY12.1* and *Pi9+Pi54+qDTY1.1+qDTY3.1*, while five IC_1_F_1_ plants with *Bph20+Bph21+qDTY2.1* were maintained as stubbles. Among 234 IC_2_F_1_ plants, four plants were identified with eight gene/QTL combinations of *xa5+xa13+Xa21+Pi9+Pi54+qDTY1.1+qDTY3.1*+*qDTY12.1*, and one to seven gene/QTLs in 25 different combinations in the remaining 230 IC_2_F_1_ plants. In the third cycle of intercrossing during *rabi* 2015, the four IC_2_F_1_ plants with eight gene/QTL combinations of *xa5+xa13+Xa21+Pi9+Pi54+qDTY1.1+qDTY3.1*+*qDTY12.1* were simultaneously advanced by selfing and crossed with five IC_1_F_1_ plants containing *Bph20+Bph21+qDTY2.1*, which were maintained as stubbles. Foreground selection in 72 IC_3_F_1_ plants identified five plants with all the target alleles *xa5+xa13+Xa21+Pi9+Pi54+Bph20+Bph21+qDTY1.1+qDTY2.1+qDTY3.1*+*qDTY12.1* and one to eight gene/QTL in myriad of combinations.

The four IC_2_F_1_ plants with eight gene/QTL combinations of *xa5+xa13+Xa21+Pi9+Pi54+qDTY1.1+qDTY3.1*+*qDTY12.1* and five IC_3_F_1_ plants with all the 11 target alleles of *xa5+xa13+Xa21+Pi9+Pi54+Bph20+Bph21+qDTY1.1+qDTY2.1 +qDTY3.1*+*qDTY12.1* were advanced by selfing up to IC_2_F_6_ and IC_3_F_6_ generations, respectively. Out of 3328 F_2_ plants from the two sets of populations, 578 single panicle selections were made and genotyped. In F_3_ generation, 49 plants were selected with five to nine target alleles in homozygous condition and grown. A total of 251 plants in F_4_ generation of both IC_2_ and IC_3_ populations were selected with desirable agronomic traits, genotyped and further advanced as families up to F_6_ generation (Figure 1).

### 2.2. Gene/QTL Introgression vis-à-vis Phenotypic Response

All the 251 IC_2_F_6_ lines were genotyped with foreground markers and phenotyped for BB, blast and BPH during *kharif* 2019 and *kharif* 2020 and for grain yield both under non-stress, as well as reproductive-stage drought stress conditions during *kharif* 2019. Out of 251 introgression lines (ILs), 196 ILs with positive phenotypic reaction to one or more targeted traits were broadly classified into seven phenotypic classes, as follows: (1) phenotypic class I with 5 ILs resistant/tolerant to BB, blast and drought; (2) phenotypic class II with 10 ILs resistant/tolerant to BB and blast; (3) phenotypic class III with 9 ILs resistant/tolerant to BB and drought; (4) phenotypic class IV with 42 ILs resistant/tolerant to blast and drought; (5) phenotypic class V with 3 ILs resistant to BB; (6) phenotypic class VI with 84 ILs resistant/tolerant to blast; and (7) phenotypic class VII with 43 ILs tolerant to drought stress (Figure 2). None of the ILs were resistant to BPH despite possessing the resistant alleles of *Bph20* and *Bph21*.

The introgression of gene/QTL in various combinations in the 196 ILs is given in Table 1, Table 2 and Table 3. Yield evaluation of all the 196 ILs under non-stress conditions and 97 ILs under reproductive-stage drought stress revealed significant phenotypic differences among the ILs, their recurrent parent—Krishna Hamsa—and four checks (Appendix A). Broad sense heritability (*H*^2^) for the grain yield was 86.00 and 70.73 under the control and reproductive-stage drought stress conditions, respectively. The descriptive statistics and critical difference (CD) at 1% and 5% level of significance (*p*-value) for days to 50% flowering and grain yield under the control and reproductive-stage drought stress are summarized in Appendix A.

#### 2.2.1. Phenotypic Class-I Resistance/Tolerance to BB, Blast and Drought

Among the five ILs, IL-19246 was resistant to BB with standard evaluation system (SES) score of 1 and blast with an SES score of 1 both during *kharif* 2019 and *kharif* 2020. The percentage yield advantage of IL-19246 over the recurrent parent Krishna Hamsa was +180% (Appendix A) with a yield of 560 g/m^2^ under non-stress conditions (Appendix A) (Figure 3). Similarly, IL-19247 was resistant to BB (SES score 1), moderately resistant to blast (SES score 4) and recorded +131% yield advantage over Krishna Hamsa under drought stress conditions and with a yield of 598 g/m^2^ under non-stress conditions. The remaining three ILs—19174, 19193 and 19196—were moderately resistant to both BB and blast with yield advantage of 82%, 68% and 120% over Krishna Hamsa under reproductive-stage drought stress (Appendix A) and a yield of 737, 374 and 544 g/m^2^ under non-stress conditions, respectively (Appendix A). Interestingly, only one IL-19196 has at least one gene *Xa21* for BB and *Pi9* for blast and two QTL-*qDTY2.1+qDTY3.1* for drought (Appendix A). Resistant alleles for blast were not observed in ILs 19174 and 19193, while resistant alleles of both BB and blast were absent in ILs 19246 and 19247 (Table 1).

#### 2.2.2. Phenotypic Class-II Resistance/Tolerance to BB and Blast

Among the 10 ILs, IL-19378 with *xa5+Xa21+Pi54*, IL 19031 with *xa5+Xa21+Bph20+Bph21*, IL 19030 with *Pi54+Bph20+Bph21* and two ILs 19019 and 19471 with *xa5+Pi9* were resistant to both BB and blast (Table 1) (Figure 4b and Figure 5b,c). IL-19030 has resistant alleles *Pi54* for blast but no resistant alleles for BB (Table 1), and IL19031 has resistant alleles for BB but not for blast. Except for *xa5* for BB resistance, none of the blast-R genes were introgressed in the remaining five ILs in this group *viz.*, ILs-19007, 19020, 19025, 19406 and 19039, but they were seen with *Bph20+Bph21* introgressions. Grain yield varied from 311 in IL 19406 to 1010 g/m^2^ in IL 19030 under non-stress conditions (Appendix A).

#### 2.2.3. Phenotypic Class-III Resistance/Tolerance to BB and Drought

The target resistant alleles of BB were missing in all except IL 19233, 19245 and 19248, and all 10 ILs were resistant to BB (SES score 1) (Figure 4c–f) and recorded +30 to +232% yield advantage over Krishna Hamsa under reproductive-stage drought stress (Appendix A) and 395 to 693 g/m^2^ of grain yield under non-stress conditions (Appendix A). ILs 19241, 19248, 19238 19240, 19232 and 19239 recorded high yield and ILs 19245, 19244 and 19233 recorded comparable yield levels with Krishna Hamsa (Appendix A).

#### 2.2.4. Phenotypic Class-IV Resistance/Tolerance to Blast and Drought

A total of 42 ILs with one to five introgressions of three genes (*xa5, Xa21* and *Pi9*) and four QTLs (*qDTY1.1, qDTY2.1, qDTY3.1* and *qDTY12.1*) in 25 different combinations were resistant to blast and tolerant to drought (Table 1) (Figure 4g,h). In as many as 19 ILs, resistant alleles of BB were without positive phenotypic reaction, while 17 ILs without resistant allele of blast were with positive phenotype to blast disease (Table 1). In this ILs, percentage yield advantage over Krishna Hamsa under reproductive drought stress varied from +14.88% to +261% (Appendix A) and yield levels varied from 180 to 1736 g/m^2^ under non-stress conditions with 19 ILs viz., 19185, 19211, 19249, 19273, 19221, 19191, 19201, 19267, 19263, 19237, 19176, 19195, 19274, 19178, 19264, 19214, 19271, 19250 and 19189, showing a higher yield than Krishna Hamsa under non-stress conditions (Appendix A).

#### 2.2.5. Phenotypic Class-V Resistance/Tolerance to BB

All the three ILs were resistant to BB with SES score of 1; however, only two ILs—19379 (Figure 5a) and 19460—have resistant alleles of *Xa21* for BB along with resistant alleles of *Pi54* for blast. None of the target resistant alleles of BB were seen in IL-19046; instead, *Bph20* and *Bph21* were introgressed in them (Table 2). Except IL-19379, the other two ILs with resistance to BB recorded a high grain yield of 2244 and 631 g/m^2^ in ILs 19046 and 19460, respectively (Appendix A). 

#### 2.2.6. Phenotypic Class-VI Resistance/Tolerance to Blast

In this class, 84 ILs were resistant to blast with one to five introgressions of 10 gene/QTLs (*xa5, xa13, Xa21, Pi9, Pi54, Bph20, Bph21, qDTY2.1, qDTY3.1* and *qDTY12.1*) combined in 31 different combinations (Table 2) (Figure 6). Out of 83 ILs with resistance to BB, 38 ILs recorded a high yield (>500 g/m2 to 2998 g/m^2^) and 7 ILs recorded comparable yield levels with Krishna Hamsa under non-stress conditions (Appendix A).

#### 2.2.7. Phenotypic Class-VII Tolerance to Drought

The percentage yield advantage in 43 ILs ranged from 23.47% to 276% over the recurrent parent Krishna Hamsa under reproductive-stage drought stress (Appendix A) and yield levels of 282 to 1282 g/m^2^ under non-stress conditions (Appendix A). Introgression of one to five gene/QTLs in 18 different combinations of three genes—*xa5, Xa21* and *Pi9*, and three QTLs*—qDTY2.1, qDTY3.1* and *qDTY12.1*, was observed in these ILs (Table 3). Among the 43 drought-tolerant ILs, 31 ILs recorded high yield and 6 ILs recorded comparable yield with Krishna Hamsa (Appendix A).

### 2.3. Background Selection

Parental polymorphism between pairs of Krishna Hamsa and each of the six donors revealed 26 to 58 polymorphic markers out of the 687 SSRs screened between them with a total of 124 polymorphic markers (Appendix A). A subset of 27 ILs viz., 19202, 19206, 19211, 19019, 19020, 19021, 19022, 19023, 19024, 19026, 19027, 19181, 19182, 19185, 19198, 19247, 19396, 19030, 19241, 19471, 19002, 19004, 19009, 19016, 19017, 19025 and 19186 with days to 50% flowering in the range of 88 to 100 days, plant height of 81 to 87 cm and a long slender grain type similar to Krishna Hamsa were selected and screened with polymorphic background markers. Background selection (BGS) revealed 73.32% to 96.43% recovery of the RP genome in the subset.

## 3. Discussion

Several high yielding crop varieties have been developed in the past using conventional breeding approaches but combining high yield with multiple stress-resistant/tolerance using this approach is tedious. Use of molecular markers not only enables trait(s) introgression from multiple donors into a single background, but also ensures retaining desirable agronomic traits of the elite recurrent parent by way of marker-assisted breeding (MAB). There are a few recent reports about effective utilization of marker-assisted breeding for the improvement of multiple stress tolerance in the background of the varieties Lalat [5], Naveen [4], Swarna [1] and ASD16 and ADT45 [43]. In the current study, a simultaneous but selective intercrossing strategy, assisted by gene-specific and tightly linked markers for target genes and peak and flanking markers of the QTLs, was deployed to combine multiple biotic and abiotic stress resistance into the genetic background of the elite rice cultivar, Krishna Hamsa. 

We employed both molecular and conventional breeding strategies in the development of multiple stress tolerance. Selective intercrossing of multiparent-derived F_1_’s was adopted in stacking multiple genes/QTL into a single background assisted by foreground selection (FGS) up to IC_3_F_1_ followed by stringent phenotypic selection for the desirable agronomic traits in the later segregating generations. Accumulating maximum gene/QTL introgressions in a common background was the main challenge in the entire process, and several hundreds of selective intercrosses in pairs were attempted from 2013 to 2015 in five consecutive seasons. Intercrossing has its own limitations, as it led to constant re-shuffling of the earlier achieved gene/QTL combinations. However, large populations were generated, which enabled the efficient selection of phenotypically acceptable plant type in the populations.

The widespread prevalence of numerous genetically distinct virulent *Xoo* strains demanded pyramiding of multiple BB-resistant genes, which can provide broad-spectrum, durable resistance in BB-prone rice growing areas. An additional BB-R gene, *Xa33*, was introgressed into Improved Samba Mahsuri possessing *xa5+xa13+Xa21* to enhance and provide broad-spectrum resistance to BB and has been released as ‘DRR Dhan 53’ for cultivation in BB endemic areas of India [7]. Additionally, recently the ICAR-Indian Institute of Rice Research (ICAR-IIRR) released an improved version of ‘Akshaydhan’ as ‘DRR Dhan 58’ introgressed with a single BB-R gene, *Xa33*, as a cultivar. Among the several resistant genes identified for blast, *Pi2, Pi9* and *Pi54* were mostly used in the development of blast-resistant cultivars. Improved versions of ‘Swarna’ with intorgession of *Pi2* have been released as ‘DRR Dhan 51’ for cultivation [29]. Similarly, in the present study, a combination of three BB-R genes, two blast-R genes, two BPH-R genes and four drought-tolerant QTLs was targeted, but we observed a positive phenotypic response to both BB and blast in t2 ILs, BB in 9 ILs and blast in 64 ILs despite the absence of corresponding R genes. Inheritance of resistance to BB and/or blast in ILs without corresponding R genes could be due to the unknown genes from other parents of the breeding scheme. Genes other than those targeted could be responsible for the tolerance, as multiple parents were involved in the development of ILs. For instance, IRBB60 was used as donor for BB genes, but it also has moderate resistance to blast [44]. Sometimes resistance pathways of different stress genes may influence each other in phenotypic expression. Expression profiling of stress-inducible genes was carried out in rice varieties and reported variability in gene expression patterns indicating the complex network of pathways for regulation of multiple stresses [45]. Detailed studies on crosstalk between defense pathways are essential [46] when genes are pyramided to engineer plants resistant to multiple stress conditions [47]. 

In the present study, we observed a negative phenotypic response in 9 ILs that were marker positive to blast-R genes, 9 ILs harboring QTLs for drought tolerance, 44 ILs that were marker positive to BPH-R genes and 67 ILs that were marker positive to BB-R genes. Significant variation was observed in the phenotypic response of the ILs in the background of Swarna despite the presence/absence of corresponding R genes and BPH susceptibility in ILs with *Bph3* and *Bph17*, which could be due to the difference in the genome recovery and background interaction of genes/QTLs of the ILs [1]. Negative interaction is specific to the recurrent parent background ‘Triguna’ and might be due to still unidentified modifier gene/s that affect the expression of resistance in plants, having the gene combinations of either *Xa21, xa13* and *xa5* or *Xa21* and *xa5* [48]. Lines with single gene introgression of *Bph20, Bph21, Bph3 or Bph18* in the background of rice maintainer line ‘Jin 23B’ had a moderate susceptible level [32]. The real effects of genes cannot be compared using the original source genotypes due to the diverse genetic backgrounds [49]. The negative phenotype in marker positive plants in our study can be the effect of varying expression of a specific gene in different background genomes. Many of these genes or QTLs were identified in different varietal or subspecies backgrounds and will not always show a consistent performance in every background; similarly, the varying pest biotypes, pathogen strains or stress load also can explain the negative phenotypic expression. Interestingly, it was observed in our lines that there is a selective exclusion of BPH resistance genes especially when BB genes are present. However, such interactions are not observed with *Pi* genes, and this requires further validation in different backgrounds to confirm the results. Multiple genes introgression in single background may result in the selective combination of compatible genes or selective exclusion of some combinations due to recombination events or chromosome-related factors [50].

We targeted four QTLs—*qDTY1.1, qDTY2.1, qDTY3.1* and *qDTY12.1*—for reproductive-stage drought tolerance. In previous studies, the introgression of these four QTLs in different combination along with *Sub1* for submergence tolerance resulted in development and release of drought- and submergence-tolerant versions of mega varieties like Swarna as CR dhan 801 (*qDTY1.1 +qDTY2.1 +qDTY3.1 +Sub1*) and CR Dhan 802 (*qDTY2.1+qDTY3.1+Sub1*) and Samba Mahsuri as DRR Dhan 50 (*qDTY2.1+qDTY3.1+Sub1*) as new cultivars in various rice growing countries of Southern Asia [41,51,52,53]. In the present study, we identified five ILs viz., IL 19,196 with *Xa21+Pi9+qDTY3.1*, ILs 19,174 and 19,193 with *xa5+Xa21+qDTY3.1* and ILs 19,246 and 19,247 with *qDTY2.1* introgressions and yield advantage of 131 to 346 g/m^2^ under reproductive-stage drought stress over Krishna Hamsa, accounting to a percentage yield advantage of 68%–180% along with BB and blast resistance. Additionally, *qDTY2.1* in combination with BB resistance resulted in a yield advantage under reproductive-stage drought stress in nine ILs in the range of 59–446 g/m^2^ (30–282%). Four QTLs—*qDTY1.1, qDTY2.1, qDTY3.1* and *qDTY12.1*—in different combinations along with blast resistance in 42 ILs resulted in the yield advantage of 29–502 g/m^2^ (14–261%) over Krishna Hamsa under reproductive-stage drought stress.

Generally, crosses involving landraces or wild species of distant gene pools are difficult due to delayed flowering or no flowering in *rabi* season. Consideration of elite donor background helped in attempting several hundreds of intercrosses in both the seasons (i.e., twice a year). Using the same recurrent parent in multiple crosses, selective intercrossing with intensive genotypic/phenotypic selection and morphological similarity among recurrent parent and donors has led to the selection of ILs with close proximity to Krishna Hamsa despite using multiple donors and not following the typical marker-assisted backcross breeding strategy. In the present study, a subset of 27 ILs were similar to Krishna Hamsa in plant type features and the data are supported retrospectively by BGS. Stacking of multiple QTLs and genes in one step with the use of a simultaneous crossing program coupled with marker-assisted forward breeding is an effective breeding approach using the elite donors used in the crossing program [5]. 

Resistance to two or more biotic and abiotic traits was achieved earlier by sequential introgressions. In the background of Improved Samba Mahsuri with inherent *xa5, xa13* and *Xa21*, DRR Dhan 58, DRR Dhan 60 and DRR Dhan 62 were developed and released as cultivars with introgression of *Saltol* QTL for salinity tolerance, *Pup1* QTL for low soil P tolerance and *Pi2* and *Pi54* for blast resistance, respectively [6,18]. The present study successfully illustrates simultaneous introgression of BB and blast resistance and reproductive-stage drought tolerance, adopting selective intercrossing assisted by foreground selection and stringent phenotyping. Similar to our findings, there are successful examples of simultaneous introgression of multiple QTL and genes for biotic and abiotic stress in rice like blast, BB, gall midge and drought tolerance in the background of Swarna [1] and Naveen [4], as well as blast, BB and drought tolerance in Lalat [5].

In the present study, as mentioned earlier, we observed a negative phenotypic response in a total of 85 ILs marker positive to resistant/tolerant gene/QTLs (67 BB+9 blast + 9 drought). However, we also witnessed a positive phenotypic response to BB and/or blast in as many as 75 ILs despite the absence of the target-resistant allele for the corresponding genes. Marker-assisted selection enables rapid introgression of targeted gene/QTLs and recovery of recurrent parent genome but does not always ensure a positive phenotypic response. Marker-assisted selection-derived genotypes resulting from the breeding strategy involving multiple parents are not always a true reflection of the targeted phenotype due to complex genic interactions and recombination events. Genome shuffling and complex recombination events due to repeated cycles of intercrossing result in an altered phenotype. Our findings suggest the need of stringent phenotyping for the targeted traits and not relying only on FGS, even more so when multiple parents are involved. Stringent phenotyping of all the lines is imperative irrespective of the targeted gene introgression when multiple donors are involved in the breeding scheme. It is also equally important to validate the marker–trait associations for each marker before venturing into such studies. The genotypes other than the donors of targeted genes may harbor unknown resistance genes, which could be the possible cause of inheritance of resistance in the ILs. 

The present study demonstrates that simultaneous introgression to multiple biotic and abiotic stress resistance/tolerance can be achieved following marker-assisted selective intercrossing among multiple single cross F1’s and their intercrossed F1’s, followed by forward breeding coupled with stringent phenotyping for the target traits. Maintenance of many plants and progenies in the early segregating generations and phenotypic selection for desirable agronomic traits enabled us to get all combinations of a positive phenotype for target traits with superior plant types. Six promising ILs—IL 19,246 with resistance to BB, blast and tolerance to drought stress, IL 19,241 with resistance to BB and drought, ILs 19,471 and 19,378 with resistance to BB and blast, IL 19,177 with resistance to blast and tolerance to drought, and IL 19001 with resistance to blast—are currently under evaluation in All India Coordinated Improvement Project (AICRIP) trials. Furthermore, several other promising ILs in various phenotypic classes from the present study will be nominated to AICRIP and once available for cultivation will benefit the farmers of boro areas. The 196 ILs from the present study falling into seven major phenotypic classes serve as a repertoire of genetic resource to choose from the myriad of gene/QTL combinations to combine resistance to more multiple abiotic and biotic stresses. The use of these ILs in future breeding programs owing to their elite cultivar background and preferred plant type traits will enable in avoiding background noise and undesirable linkage drag. 

## 4. Materials and Methods

### 4.1. Plant Material and Introgression Scheme

The present investigation was taken up at the ICAR-Indian Institute of Rice Research (ICAR-IIRR), Rajendranagar, Hyderabad, India. In the introgression scheme, the elite recurrent parent chosen was the variety, ‘Krishna Hamsa’, which was released in 1998 for cultivation in Andhra Pradesh, Tripura, West Bengal and Bihar states of India. It has a mid-early duration (115 to 120 days) and is suitable for cultivation both in *kharif* (i.e., wet season) and *rabi* (i.e., dry season) and is tolerant to low temperatures during the vegetative phase; hence, it is suitable for the highly productive Boro season in the Eastern and northeastern states of India. It also has very good grain quality and fetches good price in the local markets. 

Initially, Krishna Hamsa was simultaneously crossed with six different donors in *kharif*, 2013. (1) IRBB60 with *xa5, xa13* and *Xa21* was used as a source of BB resistance. (2) Tetep possessing *Pi9* and *Pi54* was used as a source of blast resistance. (3) IR 71033-121-15-B possessing *Bph20* and *Bph21* was used as a donor for introgression of BPH resistance. (4) IR 96,321-1447-561-B-1 with *qDTY_1.1_* and *qDTY_3.1_*, (5) IR 81,896-96-B-B-195 with *qDTY_2.1_* and (6) IR 74,371-46-1-1-13 with *qDTY_12.1_* were used as sources for incorporating drought tolerance in the background of Krishna Hamsa. In *rabi* 2014, intercrosses were attempted in three pairs of single cross F1’s. Two more cycles of selective intercrossing were attempted in the subsequent seasons by selecting sets of intercrossed F1’s with different gene/QTL combinations up to *kharif* 2015 to generate IC_2_F_1_ and IC_3_F_1_ progenies. The IC_2_F_1_ and IC_3_F_1_ progenies with maximum number of targeted gene/QTLs in various combinations were advanced further by selfing. At every generation of crossing and selfing, individual plants were genotyped using tightly linked foreground markers for the target alleles for selecting plants possessing maximum gene/QTL combinations. Stringent phenotypic selection for desirable agronomic traits was performed from IC_2_F_2_ to IC_2_F_6_ and IC_3_F_2_ to IC_3_F_6_. In IC_2_F_6_ and IC_3_F_6_ generations, lines were both genotyped and phenotyped for target traits. The introgression scheme is graphically represented in Figure 1. 

### 4.2. Genotyping

Genomic DNA was extracted by the cetyl trimethyl ammonium bromide (CTAB) method [54] from fresh leaf samples collected for all the donor and recipient parents and progenies in each generation at 2 weeks after transplanting to check for the presence of targeted gene/QTLs. For all four *qDTY* varieties (*qDTY1.1*, *qDTY2.1, qDTY3.1* and *qDTY12.1*) and eight genes (*xa5, xa13, Xa21*, *Pi2, Pi9, Pi54, Bph20* and *Bph21*), polymerase chain reaction (PCR)-based genotyping was performed. Both peak and flanking markers were used for QTL, while gene-specific and linked markers were used for genes. Here, *xa5, xa13, Xa21* and *Pi9* are functional markers, whereas *Bph20* and *Bph21* are gene-linked markers. QTL and genes were surveyed between recurrent parent and donors, and the markers with clearly distinguishable polymorphism between them were selected for foreground selection (FGS) (Appendix A). Additionally, a set of 687 SSRs spread across the rice genome were surveyed among the parents to identify polymorphic markers for use in background selection (BGS).

PCR reaction was performed in a total volume of 10 μL containing 50 ng of DNA template, 1 μL 10XPCR buffer, 2.5 picoM of each forward and reverse primer, 75 μM of each dNTP and 0.5U of *Taq* DNA polymerase (Geneilabs, India). The PCR amplification cycle was performed based on standardized annealing temperatures specific to each marker representing the gene/QTL. Products were resolved in a 3.0% agarose gel stained with EtBr and the gel images were captured using Gel documentation unit).

### 4.3. Phenotyping

#### 4.3.1. Bacterial Leaf Blight Screening

The ILs were evaluated against BB resistance during *kharif* 2019 and 2020 both under field conditions and glass house conditions using the artificial clip inoculation method along with Krishna Hamsa, Improved Samba Mahsuri (resistant check) and TN1 (susceptible check). A highly virulent, local isolate of BB pathogen, *Xanthomonas oryzae* pv. oryzae (*Xoo*) IX-020 maintained at ICAR-IIRR, was used for screening the ILs under field conditions at two locations, the ICAR-IIRR farm, Rajendranagar, Hyderabad, Telangana State, India and the International Crop Research Institute for Semi-Arid Tropics (ICRISAT), Ramachandrapuram, Hyderabad, Telangana State, India in *kharif* 2019 and the ICAR-IIRR farm, Rajendranagar in 2020, while glass house screening was done at the IIRR farm, Rajendranagar. Under field conditions, leaf tips of three plants in each IL were cut with scissors dipped in a BB suspension of 10^9^ cfu/mL at 40 days after transplanting (DAT), coinciding with the maximum tillering stage. Similarly, under glass house conditions, 45–50-day-old seedlings were inoculated. Inoculated plants were scored at 20 days after inoculation (DAI) on a 0–9 SES scale according to the Standard Evaluation System, IRRI [55], based on lesion length measured from the cut end of the leaf. A plant was classified as resistant if the average lesion length was shorter than 3 cm, moderately resistant if the lesion was >3–6 cm, moderately susceptible if the lesion was >6–9 cm and susceptible if the lesion was longer than >9 cm [7].

#### 4.3.2. Blast Screening

Blast screening was performed in universal blast nursery (UBN) facility at ICAR-IIRR, Rajendranagar during *kharif* 2019 and *kharif* 2020. In raised nursery beds with a row spacing of 10 cm, one row of the susceptible check (HR-12) was planted between every four entries and also along the borders to facilitate the build-up of inoculum for uniform and rapid spread of the disease. The inoculums with the concentration of 1 × 10^5^ spore/mL were sprayed onto young seedlings at four leaf stages using fine sprayer, and high relative humidity was maintained for disease development. Tetep was used as resistant check, and plants were scored for blast disease reaction at 20 DAI on a 0–9 SES scale IRRI [53]. SES scores of 0–3 were considered resistant (R), 4–5 as moderately resistant (MR) and 6–9 as susceptible. 

#### 4.3.3. Brown Plant Hopper Screening

The seedling box technique was used for screening of ILs against BPH during *kharif* 2018 at ICAR-IIRR, Rajendranagar. BPH biotype 4 from IIRR Rajendranagar was used to screen BPH under controlled glass house conditions. Newly hatched nymphs or adults were utilized for screening ILs during *kharif* 2018 along with recurrent parents and TN1 and PTB33 as susceptible (S) and resistant (R) checks, respectively, following the standard protocol [56]. Seedlings were sown in a tray in three replications with two border rows of the S check and one row of R check in the center. Seedlings of 3–4 leaf stage at 10 days after sowing were infested with 6–8 instars nymphs per seedling, and the evaluation of damage was based on SES scores on a scale of 0–9 [57].

#### 4.3.4. Yield Evaluation under Non-Stress and Drought Stress

The ILs were evaluated for their agronomic performance both under control and drought stress situations during *kharif* 2019 at the IIRR farm, ICRISAT. For the non-stress experiment, the ILs along with their recurrent parents were sown in raised nursery beds on 19 June 2019. Each line was sown in a 1 m row length with spacing of 10 cm between the rows. Seedlings at the age of 31 days were transplanted to the main field on 13 July 2019. Since the drought trial in 2019 was conducted during *kharif*, sowing and transplanting were delayed by 20 days to prevent the crop experiencing rainy days during reproductive stage. The standard agronomic package of practices was followed while growing the rice plants. Irrigation was supplied continuously in a non-stress experiment until maturity. In drought stress trial, reproductive-stage drought stress was imposed by withholding the irrigation at 30 DAT and draining out the remaining water in the field. Perforated PVC pipes of about 1 m length were inserted diagonally in the drought trial plot to monitor the below ground soil moisture content. The decline in water table depth was measured on a daily basis with a meter scale inserted into the PVC pipes. Lifesaving irrigation was provided for 24 h when the water table level reached 100 cm below the soil surface and most lines were wilted and exhibited severe leaf drying. Then, a second cycle of the stress was initiated that continued until maturity. 

In both the trials, the experiment was laid out in an augmented randomized complete block design with repetition of Krishna Hamsa, MTU1010, IR 64 and DRR Dhan 44 as checks. Each entry was planted in two rows with a row length of 3.45 m following the spacing of 20 cm between the rows and 15 cm between the hills. The experimental material was evaluated for days to 50% flowering and grain yield in g/m^2^.

### 4.4. Statistical Analysis

Yield data from non-stress and drought stress trials were subjected to statistical analysis using R Studio (Version 3.5.2, R Core Team, Vienna, Austria) using R-scripts. Analysis of variance (ANOVA) and critical difference (CD) at the 1% and 5% level of significance (*p*-value) were used for assessing the variance and testing of significant differences among the ILs, respectively, and to understand the variation and descriptive statistics, the broad sense heritability (H^2^) and F test were calculated. CD, representing confidence intervals of all levels for a parameter of interest, was calculated using CD = SE (d) × t, where S.E (d) = √EMS/r and t is the critical value for a specified level of significance and error degrees of freedom. Heritability in broad sense was computed by using the following:$$h^{2}=\frac{\sigma^{2}g}{\sigma^{2}p}\times 100$$
where,

*h*^2^ = Heritability (broad sense);

*σ* ^2^*_g_* = Genotypic variance; 

*σ* ^2^*_p_* = Phenotypic variance. 

## Figures and Tables

**Figure 1 plants-11-00622-f001:**
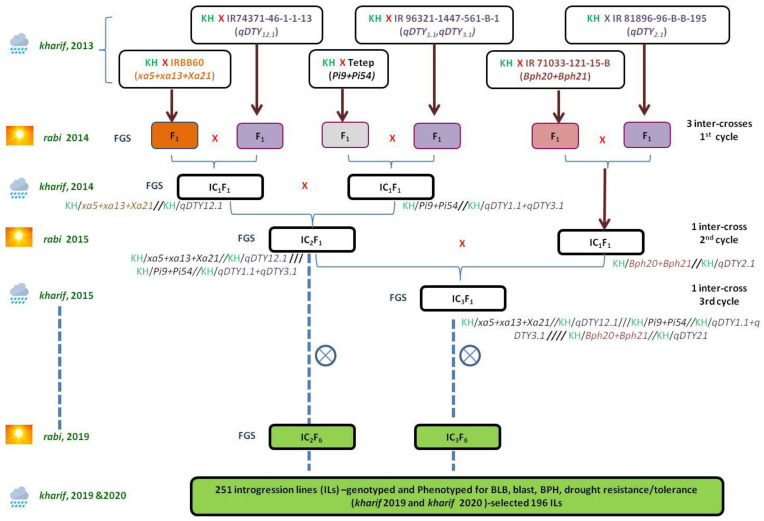
Introgression scheme involving elite recurrent parent, ‘Krishna Hamsa’, and six donor parents for biotic and abiotic traits. IRBB 60 with *xa5+xa13+Xa21* for BB, Tetep with *Pi9* for blast and IR 71033-121-15-B with *Bph20+Bph21* for BPH genes, as well as IR 96321-1447-561-B-1 with *qDTY1.1+qDTY3.1*, IR 81896-96-B-B-195 with *qDTY2.1* and IR74371-46-1-1-13 with *qDTY12.1* for drought-tolerant QTLs. *kharif* is the wet season with crop growing period from June to November, and *rabi* is the dry season with crop growing period from December to May.

**Figure 2 plants-11-00622-f002:**
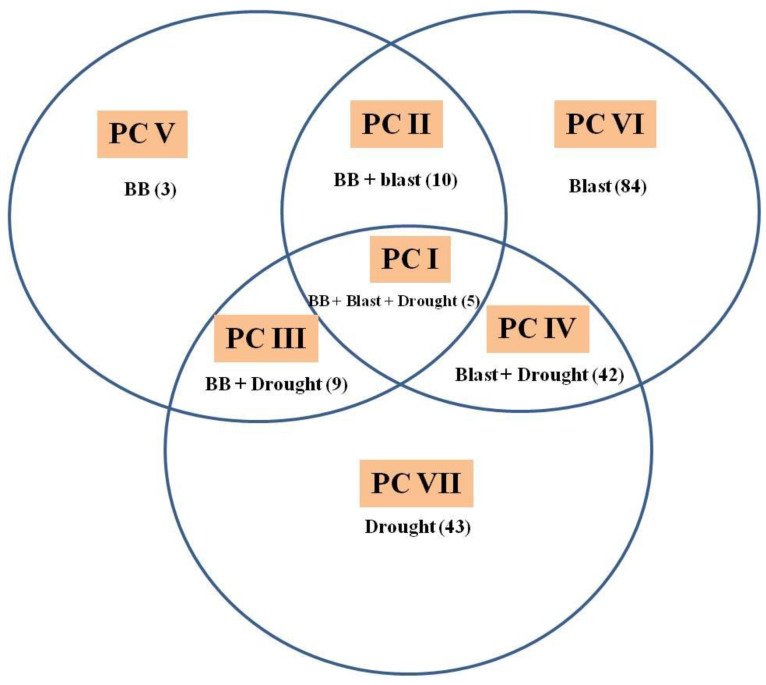
Venn diagram depicting resistance/tolerance of 196 introgression lines (ILs) in seven major phenotypic classes (PC) of bacterial blight (BB), blast and drought in various combinations. Phenotypic classes are represented as PC-I to PC-VII. Number in parenthesis represents the total ILs in that PC.

**Figure 3 plants-11-00622-f003:**
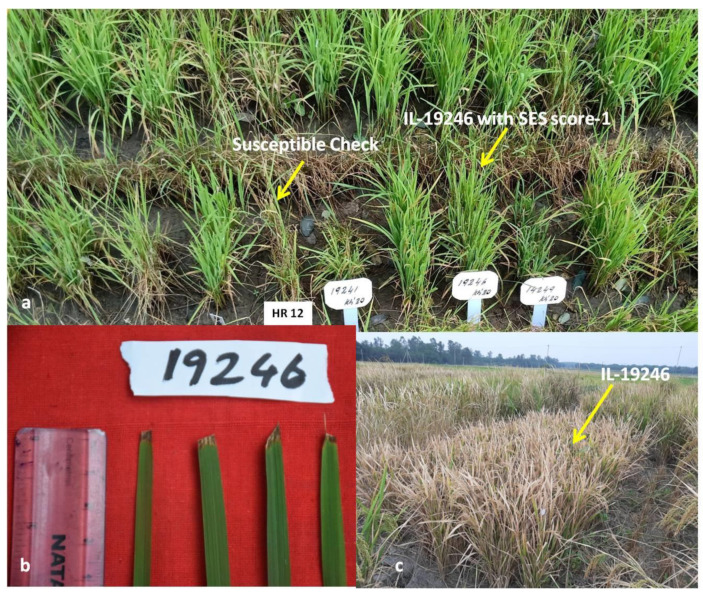
Performance of the IL-19246 of phenotypic class-I with resistance to blast in the universal blast nursery (UBN) during *kharif* 2019 and *kharif* 2020 and resistance to BB in glass house and field screening during *kharif* 2019 and *kharif* 2020, as well as tolerance to drought during *kharif* 2019. IL-19246 with (**a**) blast resistance score of 1 as per standard evaluation system (SES) in UBN during *kharif* 2020. ‘HR12’ was used as susceptible check (SES score 9). (**b**) BB resistance (SES score 1) during *kharif* 2019 in glass house screening and (**c**) yield of 538 g/m^2^ under reproductive-stage drought stress conditions with +180% yield advantage over ‘Krishna Hamsa’ during *kharif* 2019.

**Figure 4 plants-11-00622-f004:**
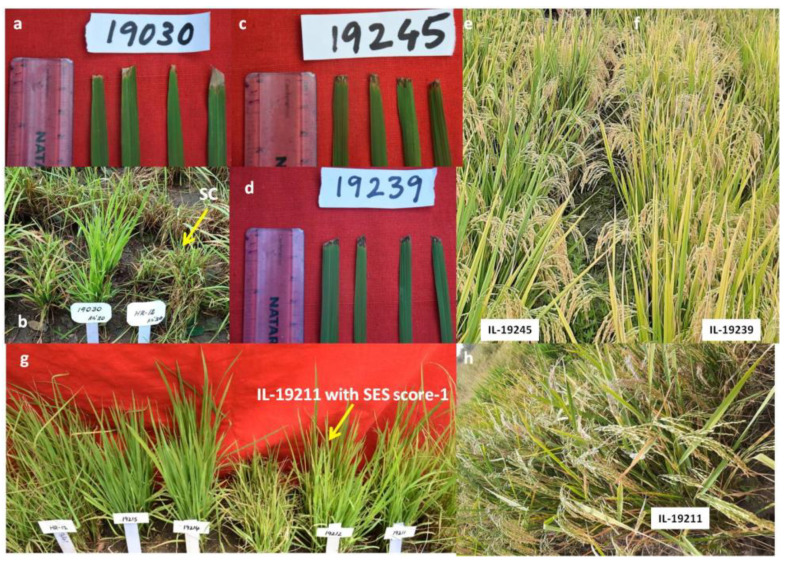
Performance of the ILs of phenotypic classes-II, -III and -IV with resistance/tolerance to two traits. (**a**,**b**): IL-19030 of PC-II with BB resistance (SES score 1) during *kharif* 2019 in glass house screening and blast resistance (SES score 1) in UBN during *kharif* 2020. (**c**–**f**): ILs 19245 and 19239 of PC-III with BB resistance (SES score 1) during *kharif* 2019 and yield advantage of +231 and +59.9% over Krishna Hamsa under reproductive-stage drought stress during *kharif* 2019. (**g**,**h**): IL 19211 of PC-IV with blast resistance (SES score 3) and +135% yield advantage over Krishna Hamsa under reproductive-stage drought stress during *kharif* 2019.

**Figure 5 plants-11-00622-f005:**
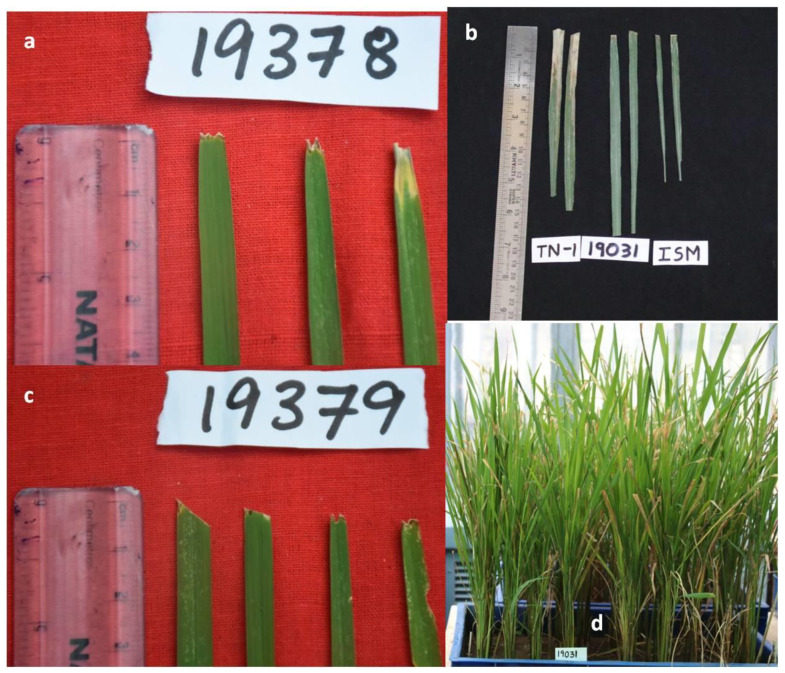
Performance of the ILs-19,378, 19,379 and 19,031 of phenotypic class-V with resistance to BB in glass house and field screening during *kharif* 2019 and *kharif* 2020. ILs can be seen with immune reaction to BB. BB resistance with SES score-1 in (**a**) IL-19,378 during *kharif* 2019 glass house screening, (**b**) IL-19,031 during *kharif* 2020 field screening, (**c**) IL-19,379 during *kharif* 2019 glass house screening and (**d**) IL-19031 during *kharif* 2020 glass house screening. TN1 was used as a susceptible check and Improved Samba Mahsuri as a resistant check for BB disease.

**Figure 6 plants-11-00622-f006:**
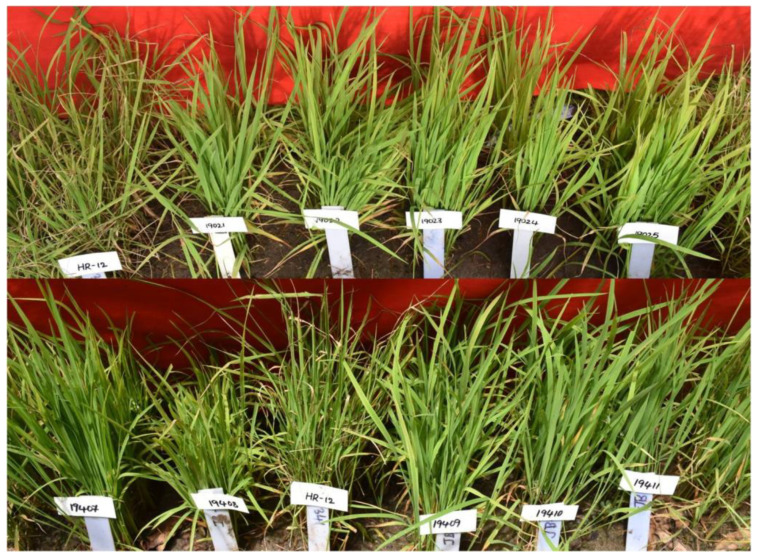
Performance of the ILs of PC-VI with resistance to blast in universal blast nursery (UBN) during *kharif* 2019 and *kharif* 2020. ‘HR-12’ susceptible check (SC) with SES score of 9, and ILs 19021 to 19025 and 19407 to 19411 with blast resistance (SES score—1) during *kharif* 2019.

**Table 1 plants-11-00622-t001:** Gene/QTL introgressions in the introgression lines (ILs) of phenotypic classes with resistance/tolerance to three (BB + blast + drought) and two (BB + blast; BB + drought; blast + drought) traits.

Gene/QTL Combination	ILs with BB, BL Scores and % YA over RP under RDS	No. of ILs
Phenotypic class I: Resistance/tolerance to BB, blast and drought (three traits)		
*Xa21+Pi9+qDTY2.1+qDTY3.1*	19196 (5, 4, +120.27%)	1
*xa5+Xa21+qDTY3.1*	19174 (5, 4, +82.28%), 19193 (5, 4, +68.23%)	2
*qDTY2.1*	19246 (1, 1, +180.43%), 19247 (1, 4, +131.13%)	2
Phenotypic class II: Resistance/tolerance to BB and blast (two traits)		
*Pi54+Bph20+Bph21*	19030 (1,1)	1
*xa5+Bph20+Bph21*	19007 (5, 3), 19020 (5, 1), 19025 (5, 1), 19039 (5, 4), 19406 (5,1)	5
*xa5+Pi9*	19019 (3,1), 19471 (3,1)	2
*xa5+Xa21+Bph20+Bph21*	19031 (1,4)	1
*xa13+Xa21+Pi54*	19378 (1,1)	1
Phenotypic class III: Resistance/tolerance to BB and drought (two traits)		
*Pi9+qDTY2.1*	19232 (1, +127.06%), 19238 (1, +73.23%), 19239 (1, +59.66%), 19240 (1, +125.52%), 19244 (1, +30.72%)	5
*xa5+Pi9+qDTY2.1*	19233 (1, +232.44%)	1
*qDTY2.1*	19241 (1, +60.12%),	1
*xa5+qDTY2.1*	19245 (1, +230.63%), 19248 (1, +135.20%)	2
Phenotypic class IV: Resistance/tolerance to blast and drought (two traits)		
*xa5+Xa21+Pi9+qDTY12.1+qDTY2.1*	19208 (1, +72.78%)	1
*xa5+Xa21+Pi9+qDTY12.1+qDTY3.1*	19182 (3, +46.09%)	1
*xa5+Xa21+Pi9+qDTY2.1+qDTY3.1*	19194 (4, +101.56%), 19195 (4, +153.65%), 19197 (4, +55.14%), 19198 (4, +260.03%), 19200 (4, +98.11%), 19201 (4, +198.97%), 19267 (3, +164.60%)	7
*xa5+Xa21+qDTY1.1+qDTY2.1+qDTY12.1*	19177 (5, +55.14%)	1
*Xa21+Pi9+qDTY12.1+qDTY3.1*	19185 (3, +14.88%)	1
*Xa21+Pi9+qDTY2.1+qDTY3.1*	19199 (4, +167.76%)	1
*Xa21+qDTY12.1+qDTY2.1+qDTY3.1*	19189 (4, +82.73%), 19191 (3, +80.47%), 19192 (4, +133.39%)	3
*xa5+Xa21+Pi9+qDTY12.1*	19263 (4, +129.32%)	1
*xa5+Xa21+Pi9+qDTY2.1*	19206 (3, +42.02%)	1
*qDTY12.1+qDTY2.1+qDTY3.1*	19190 (1, +68.75%)	1
*Xa21+Pi9+qDTY2.1*	19214 (2, +113.49%), 19215 (2, +211.18%), 19249 (4, +140.17%)	3
*xa21+pi9+qDTY3.1*	19237 (4, +62.83%), 19262 (4, +178.62%), 19264 (1, +155.10%), 19271 (4, +52.43%)	4
*Xa21+qDTY12.1+qDTY2.1*	19250 (4, +98.56%)	1
*Xa21+qDTY2.1+qDTY3.1*	19261 (4, +154.19%)	1
*xa5+Pi9+qDTY2.1*	19203 (3, +55.59%)	1
*xa5+qDTY2.1+qDTY3.1*	19211 (3, +134.75%)	1
*xa5+Xa21+qDTY3.1*	19176 (4, +68.71%)	1
*Pi9+qDTY3.1*	19271 (4, +52.43%), 19274 (4, +202.59%), 19279 (3, +148.31%)	3
*qDTY12.1+qDTY3.1*	19183 (4, +113.94%)	1
*qDTY2.1+qDTY3.1*	19253 (1, +126.15%), 19254 (4, +222.04%)	2
*Xa21+qDTY2.1*	19268 (4, +188.57%)	1
*Xa21+qDTY3.1*	19221 (4, +114.39%), 19275 (4, +143.34%)	2
*xa5+qDTY2.1*	19205 (4, +16.69%)	1
*qDTY12.1*	19178 (5, +256.87%)	1
*qDTY3.1*	19181 (3, +163.69%)	1

Values in parenthesis indicate mean phenotypic data of BB, blast scores and percent yield advantage over Krishna Hamsa, the recurrent parent (%YA over RP), under reproductive-stage drought stress (RDS) for ILs in PC-I, BB and blast in PC-II, BB and %YA over RP for ILs in PC-III and blast and %YA over RP for ILs in PC-IV in sequence. Standard evaluation system (SES) score of BB is the mean of four environments—phenotyping in the glass house and field during *kharif* 2019 and *kharif* 2020. SES score of blast is the mean of two environments—phenotyping in the universal blast nursery (UBN) during *kharif* 2019 and *kharif* 2020. %YA over RP under RDS is the phenotyping during *kharif* 2019.

**Table 2 plants-11-00622-t002:** Gene/QTL introgressions in the ILs of phenotypic class with resistance/tolerance to single trait (BB; blast).

Gene/QTL Combination	IL with Mean BB/Blast Score	No. of ILs
Phenotypic class V: resistance/tolerance to bacterial blight		
*xa5+Xa21+Pi54*	19460 (1)	1
*xa5+Xa21+Pi54*	19379 (1)	1
*Bph20*	19046 (1)	1
Phenotypic class VI: resistance/tolerance to blast		
*Xa21+Pi9+qDTY12.1+qDTY2.1+qDTY3.1*	19188 (1)	1
*xa5+Pi9+pi54+Bph20+Bph21*	19013 (1)	1
*Xa21+Pi9+Bph20+Bph21*	19392 (4)	1
*xa5+Pi9+Bph20+Bph21*	19464 (4)	1
*Xa21+Pi9+qDTY12.1+qDTY2.1*	19186 (1)	1
*xa5+Pi9+qDTY2.1+qDTY3.1*	19172 (1)	1
*xa5+Xa21+Pi9+qDTY2.1*	19207 (1)	1
*xa21+pi9+qDTY3.1*	19167 (1)	1
*Bph20+Bph21+Pi9*	19386 (1), 19387 (1)	2
*Pi54+Bph20+Bph21*	19001 (1), 19014 (2), 19015 (2), 19033 (1), 19035 (4), 19037 (4), 19038 (4), 19043 (4), 19044 (1), 19045 (1)	10
*xa5+Pi9+Bph20+Bph21*	19022 (1)	1
*Pi9+Bph20+Bph21*	19006 (4)	1
*xa5+Pi9+qDTY2.1*	19204 (4)	1
*Bph20+Bph21*	19004 (4), 19005 (3), 19008 (1), 19021 (1), 19026 (1), 19028 (4), 19032 (3), 19040 (3), 19394 (1), 19405 (4),	10
*xa5+Bph20+Bph21*	19023 (2), 19024 (1)	2
*xa5+qDTY2.1*	19202 (1)	1
*Pi9+qDTY3.1*	19270 (1)	1
*Pi9+qDTY3.1*	19272 (1)	1
*Pi9+Bph20*	19052 (4)	1
*Pi9+Pi54*	19055 (4)	1
*xa13+Pi9*	19396 (2), 19401 (3), 19402 (3), 19411 (2), 19413 (4), 19467 (5), 19470 (3)	7
*Xa21+Pi54*	19459 (5)	1
*xa5+Pi9*	19399 (4), 19407 (3), 19408 (3), 19409 (2), 19461 (3)	5
*Bph20*	19048 (4), 19049 (4), 19053 (3), 19054 (4), 19056 (3)	5
*Pi54*	19027 (3), 19034 (4), 19042 (4), 19050 (4)	4
*xa5 +Pi9*	19018 (3)	1
*xa5+Pi9*	19420 (2), 19421 (2)	2
*xa13*	19389 (4), 19403 (4), 19410 (2), 19412 (1), 19415 (4), 19416 (3), 19462 (4), 19463 (3), 19465 (4)	9
*Xa21*	19180 (3), 19243 (4), 19466 (4), 19468 (4)	4
*xa5*	19397 (1), 19417 (4)	2
*xa5+xa13*	19469 (4)	1
*xa5+Xa21+Pi9*	19210 (3)	1
*No genes*	19175 (4), 19400 (1)	2

Values in parenthesis indicate mean phenotypic data of BB for ILs in PC-V and blast in PC-VI. SES score of BB is the mean of four environments—phenotyping in the glass house and field during *kharif* 2019 and *kharif* 2020. SES score of blast is the mean of two environments—phenotyping in the universal blast nursery (UBN) during *kharif* 2019 and *kharif* 2020.

**Table 3 plants-11-00622-t003:** Gene/QTL introgressions in the ILs of phenotypic class-VII with tolerance to drought.

Gene/QTL Combination	ILs with % YA over RP under RDS	No. of ILs
*xa5+Xa21+Pi9+qDTY12.1+qDTY3.1.*	19280 (+204.4%), 19281 (+149.22%)	2
*xa5+Xa21+qDTY12.1+qDTY2.1+qDTY3.1*	19184 (+49.26%)	1
*Xa21+Pi9+qDTY2.1+qDTY3.1*	19257 (+79.11%), 19258 (+115.75%), 19259 (+128.87%), 19266 (+146.05%)	4
*xa5+Pi9+qDTY12.1+qDTY2.1*	19209 (+87.71%)	1
*xa5+Xa21+Pi9+qDTY12.1*	19266 (+146.05%)	1
*Xa21+Pi9+qDTY2.1*	19222 (+164.6%), 19223 (+16.86%), 19224 (+189.93%), 19225 (+184.5%), 19231 (+215.25%), 19234 (+203.04%), 19255 (+23.93%)	7
*Xa21+pi9+qDTY3.1*	19170 (+134.29%)	1
*xa5+Pi9+qDTY2.1*	19217 (+82.73%), 19218 (+91.78%), 19219 (+172.29%), 19220 (+32.52%), 19229 (+32.52%)	5
*xa5+Xa21+qDTY12.1*	19173 (+154.65%)	1
*Pi9+qDTY2.1*	19213 (+101.73%), 19226 (+275.86%), 19227 (+63.73%), 19228 (+103.08%), 19230 (+23.48%), 19242 (+146.5%)	6
*pi9+qDTY3.1*	19168 (+26.64%), 19171 (+28.91%), 19277 (+184.95%), 19278 (+243.75%)	4
*qDTY2.1+qDTY3.1*	19166 (+110.32%)	2
*Xa21+qDTY2.1*	19235 (+161.88%), 19251 (+216.61%), 19269 (+153.65%)	3
*xa5+qDTY2.1*	19236 (+112.58%)	1
*qDTY12.1*	19187 (+43.38%)	1
*qDTY2.1*	19252 (+134.75%)	1
*qDTY3.1*	19169 (+83.63%)	1
*xa5+Xa21+qDTY2.1*	19216 (+134.75%)	1

Values in parenthesis indicate percent yield advantage over Krishna Hamsa, the recurrent parent (%YA over RP), under reproductive-stage drought stress (RDS) for ILs in PC-VII during *kharif* 2019.

## Data Availability

Not applicable.

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
