# Peer review of "Multiparent-Derived, Marker-Assisted Introgression Lines of the Elite Indian Rice Cultivar, ‘Krishna Hamsa’ Show Resistance against Bacterial Blight and Blast and Tolerance to Drought"

_plants, 2022, doi:10.3390/plants11050622_

Round 1
Reviewer 1 Report
The ms is the result of an intensive and years-long work of breeding, that eventually pyramidized several resistance genes and QTLs for abiotic traits into an indian elite rice variety. The work is not unprecedented in the concept, but provides interesting breeding material for specific indian farmers, which is an important achievement and is correctly taken up by authorities and international research centers.
The main issue with the ms is the quality of the figures: they are not clearly representing the data, arrows and text may help the reader, the controls are missing in most cases, and a general lack of precision is affecting them. Also, the legends are not clearly describing all the terms and should be revised. Legends should describe what is to be observed in the figure, possibly pointing to parts of special interest.
Explaining briefly which and what are the indian sesasons during which experiments have been conducted would help clarity.
The phenotypic data are important, but currently they are presented only as additional material. The authors should consider reshaping the tables, eliminating unnecessary columns (e.g. no. of genes / QTLs, which is already represented in column 1) and add the most striking phenotypic data, when possible and relevant for discussion. Also "entry nos." is probably not clear and ILs no should maybe considered instead.
The data on bgs are not sufficiently presented in the text nor discussed.
Minor issues:
- revise the use of "-" instead of "( )" when mentioning the numbers of the ILs, or any other system which would ensure a better homogeneity in comparison to the one present in the ms.
- check lines 21, 37, 93, 153, 183-184, 312, 351-353, 360, 377, 466
- stick to BB instead of BLB
- consider anticipating the explanation of the acronyms which are eventually presented only in the M&M, to facilitate the reading; consider this aspect also in the legends
- Lines 403 et seq. are redundant
- lines 414 et seq. seem to be contradicting lines 351 et seq.
- the 85 ILs presented at 414 seem to be a bit out of the blue: a short intro to where they are coming from would help the reader.
Author Response
Point-by-point response to the reviewer's comments are given below
- The ms is the result of an intensive and years-long work of breeding, that eventually pyramidized several resistance genes and QTLs for abiotic traits into an indian elite rice variety. The work is not unprecedented in the concept, but provides interesting breeding material for specific indian farmers, which is an important achievement and is correctly taken up by authorities and international research centers.
Response: Authors immensely thank reviewer for the constructive suggestions and valuable comments which greatly helped in the improvement of the manuscript. The corrections have been incorporated as per the suggestions made by the reviewer.
- The main issue with the ms is the quality of the figures: they are not clearly representing the data, arrows and text may help the reader, the controls are missing in most cases, and a general lack of precision is affecting them. Also, the legends are not clearly describing all the terms and should be revised. Legends should describe what is to be observed in the figure, possibly pointing to parts of special interest.
Response: Figures have been revised with addition of arrows and text to represent the data. Since a large number of ILs were screened, susceptible check (SC) was used with a set of ILs and hence SC couldn’t be shown in all figures. However, SC ‘HR12’ for blast disease was shown in all blast screening figures 3 to 5. Similarly, in Fig 5, ‘TN1’ and ‘Improved Samba Mahsuri’ as susceptible and resistant checks respectively for BB in comparison with IL-19031 were shown. Authors once again thank the reviewer for pointing out the mistake in legends. Now, we have rephrased the legends clearly describing all the terms.
- Explaining briefly which and what are the Indian seasons during which experiments have been conducted would help clarity.
Response: Figure 1 legend has been revised with inclusion of the details of kharif and rabi seasons. kharif is the wet season with crop growing period from June to November and rabi is the dry season with crop growing period from December to May. We described kharif as wet season and rabi as dry season in the materials and methods section also.
- The phenotypic data are important, but currently they are presented only as additional material. The authors should consider reshaping the tables, eliminating unnecessary columns (e.g. no. of genes / QTLs, which is already represented in column 1) and add the most striking phenotypic data, when possible and relevant for discussion. Also "entry nos." is probably not clear and ILs no should maybe considered instead.
Response: Since the number of ILs is large, presenting phenotypic data for each IL will result in increasing the size of the main tables, hence data was earlier presented in supplementary tables. As suggested by the reviewers, we have revised the tables and presented the mean phenotypic data of BB, blast and drought screening in the main tables for each IL in parenthesis. Column on ‘no. of genes/QTL’ has been removed as suggested. Entry nos have been replaced with IL No as suggested.
- The data on bgs are not sufficiently presented in the text nor discussed.
Response: The present work is not essentially a backcross breeding program aimed at development of near isogenic lines. However, Krishna Hamsa was the common background into which several genes/QTL were targeted from multiple donors and considering the morphological similarity between 27 ILs and Krishna Hamsa, background selection was done retrospectively. BGS validated our observations on morphological similarity. The same has been discussed in the 5th para under ‘discussion’. Also results on BGS have been presented under subsection 2.3 of results with data on polymorphic markers for BGS in supplementary table S8.
Minor issues:
- revise the use of "-" instead of "( )" when mentioning the numbers of the ILs, or any other system which would ensure a better homogeneity in comparison to the one present in the ms.
Response: “()” have been removed while mentioned the numbers of the ILs and sentences have been revised appropriately in the manuscript.
2. check lines 21, 37, 93, 153, 183-184, 312, 351-353, 360, 377, 466
Response: The above mentioned lines have been checked and found either spelling mistakes or revision of sentences. Accordingly, corrections were made.
3. stick to BB instead of BLB
Response: BLB has been replaced with BB throughout the manuscript.
4. consider anticipating the explanation of the acronyms which are eventually presented only in the M&M, to facilitate the reading; consider this aspect also in the legends
Response: Legends of figures and tables and text in the results section have been revised with explanation of acronyms as suggested.
5. Lines 403 et seq. are redundant
Response: The sentence here is required to maintain flow of the subsequent content.
6. lines 414 et seq. seem to be contradicting lines 351 et seq.
Response: Both lines explain our observations in different sets of ILs. Sentence at 351 explains susceptibility in ILs despite possessing the targeted gene/QTLs while sentence at 414 explains resistance in ILs despite the absence of targeted gene/QTLs.
7. the 85 ILs presented at 414 seem to be a bit out of the blue: a short intro to where they are coming from would help the reader.
Response: The 85 ILs is a sum total of nine ILs marker positive to blast- R genes, nine ILs harbouring QTLs for drought tolerance and 67 ILs marker positive to BB-R genes mentioned at the beginning of 4th paragraph of discussion. As suggested, we have added in brief about the same at line 414.
Reviewer 2 Report
Dear Editor,
Thank you for inviting me to review this manuscript. The paper itself is well written, although 1) somewhat results are partially descriptive and partially inferential. However, the authors have conducted a thorough literature review, undertaken a rigorous piece of data collection, and have generalized information accurately.
- With minor grammatical revisions, the manuscript can be accepted as is.
- It is also acknowledged that this paper is probably the first of many papers to emerge from the study. As such, it is an overview paper that raises many questions. It would be interesting for the authors to provide more information about the research design and estimation methodologies, such as chi-square analysis of introgression lines, if possible for each crossed IL population. I only recommend some minor revisions before acceptance.
- This work is of outstanding quality, and I normally present more critical points in my reviews. However, this time it is just very beautiful work.
Line 262: “=261 %” , and Line 312 “linkes” a typo? Correct it
It was a pleasure to read this manuscript. I wish the author of the best.
Author Response
Point-by-point response to reviewer's comments
- Thank you for inviting me to review this manuscript. The paper itself is well written, although 1) somewhat results are partially descriptive and partially inferential. However, the authors have conducted a thorough literature review, undertaken a rigorous piece of data collection, and have generalized information accurately.
Response: Authors profusely thank the reviewer for the appreciation.
- With minor grammatical revisions, the manuscript can be accepted as is.
Response: Yes, we agree that there were some typo and spelling mistakes in the manuscript. The manuscript has been thoroughly revised for the same.
- It is also acknowledged that this paper is probably the first of many papers to emerge from the study. As such, it is an overview paper that raises many questions. It would be interesting for the authors to provide more information about the research design and estimation methodologies, such as chi-square analysis of introgression lines, if possible for each crossed IL population. I only recommend some minor revisions before acceptance.
Response: Authors once again thank the reviewer for correct assessment of the basic purpose of the manuscript. Yes, this is truly an overview paper presenting the interesting observations from our study. The introgression scheme and phenotyping of various biotic traits and drought phenotyping have been described in detail under ‘Materials and Methods’ section. The metric data on yield traits was statistically analyzed and results of ANOVA, heritability and critical differences have been presented in the manuscript. Authors agree that it would be more inferential with chi square values. However, the crossing scheme was viewed holistically and data on each cross was not maintained separately as the present study aimed at selecting introgression lines from multiples crosses with multiple stress resistance/tolerance by pooling several genes and QTLs into a common background. Despite maintaining large base populations, plants per se were selected based on marker positivity for inter-crossing and selfing and further stringent phenotypic selection for the targeted traits. Hence, chi-square which is perfectly apt for population derived from biparental crosses is not used in our study.
- This work is of outstanding quality, and I normally present more critical points in my reviews. However, this time it is just very beautiful work.
Response: Authors feel greatly encouraged and motivated with the reviewer’s comments.
Line 262: “=261 %” , and Line 312 “linkes” a typo? Correct it
Response: ‘=261%’ has been corrected to ‘+261%’ and typo error of ‘linkes’ corrected to ‘linked’
- It was a pleasure to read this manuscript. I wish the author of the best.
Response: Thank you for the positive comments.
Reviewer 3 Report
This paper described the multi-parents introgression assisted by molecular markers. The content is informative, however, the tables should be reorganized and the statistical methods description have to be improved. Please see the suggestion and comments below.
- L48-63: should add some citations on the first part of introduction.
- L127: Please add the explanation of “boro season”.
- The legend of supplementary table should be improved. Table and main text are independent, so the authors have to describe the table more carefully.
- L562: the section of statistical analysis should add more details. For example, how does H2 calculate? Also please indicate the R version.
- On Supplementary table S2, does significant mean p-value? Indicate DFF = days to fifty percent flowering. What is the “treatment”? What do the results on “check” mean? How did you analyze “control vs IL”?
- On Supplementary table S3, similar questions as S2, please also explain.
- On Supplementary table S5, what is “C.D”? How did you calculate those values on those comparison types?
- Please re-organize the Supplementary table S5, I suggest use each ILs only appear one time and add one more column to show their PC groups. Then the table can be more informative and ease to read.
- Please add the full name of abbreviations. For example on L203 “CD”, L211 “SES“, L228 “UBN“ and L332 “ICAR-IRRR”.
- The last paragraph of results is about background selection. Suggest to add subtitle “2.2.8”.
- Please add some gel pictures of the foreground selection markers you used in order to visualize the genotyping results and showed the polymorphism of these markers on gel.
- L296-L302: the marker descriptions on main text cannot match the supplementary table S8. Also, what are those 27 ILs on L298?
- L602: there is no appendix.
Author Response
Point-by-point response to reviewer's comments
This paper described the multi-parents introgression assisted by molecular markers. The content is informative, however, the tables should be reorganized and the statistical methods description have to be improved. Please see the suggestion and comments below.
Response: We appreciate the suggestions from the reviewer, which has resulted in improving the message of the manuscript.
- L48-63: should add some citations on the first part of introduction.
Response: Added references appropriately at two places as suggested by the reviewer.
2. L127: Please add the explanation of “boro season”.
Response: Explanation for boro season has been added.
3. The legend of supplementary table should be improved. Table and main text are independent, so the authors have to describe the table more carefully.
Response: Legends of the supplementary tables have been revised and inference of the table is given in foot note.
4. L562: the section of statistical analysis should add more details. For example, how does H2calculate? Also please indicate the R version.
Response: Authors once again thank the reviewer for the valuable suggestion. More details on the statistical analysis have been added as suggested.
5. On Supplementary table S2, does significant mean p-value? Indicate DFF = days to fifty percent flowering. What is the “treatment”? What do the results on “check” mean? How did you analyze “control vs IL”?
Response: Yes, significant means p-value. In the supplementary tables, expanded form of DFF is given as suggested. Treatments refers to introgression lines and check to control. For uniformity, we have changed the terminology to treatment instead of using IL or variety and check to control in all the revised supplementary tables and rephrased the legends accordingly.
6. On Supplementary table S3, similar questions as S2, please also explain.
Response: Similar to Supplementary Table S2, S3 has been revised
7. On Supplementary table S5, what is “C.D”? How did you calculate those values on those comparison types?
Response: CD is the critical difference at 1% and 5% level of significance (p-value) for testing of significant differences among the ILs. The details of CD calculation have been added in the materials and methods section as suggested.
8. Please re-organize the Supplementary table S5, I suggest use each ILs only appear one time and add one more column to show their PC groups. Then the table can be more informative and ease to read.
Response: Supplementary table S5 is on CD and corrections have been addressed as suggested as at S. No 7. Supplementary table S6 have been modified by shifting data of BB and blast scores to main table. Each PC group is mentioned on top as sub heading at the start of each group. Each IL is presented only once in the entire table under separate PC groups.
9. Please add the full name of abbreviations. For example on L203 “CD”, L211 “SES“, L228 “UBN“ and L332 “ICAR-IRRR”.
Response: Full names of the abbreviations have been added as suggested.
10. The last paragraph of results is about background selection. Suggest to add subtitle “2.2.8”.
Response: Subtitle has been added as suggested.
11. Please add some gel pictures of the foreground selection markers you used in order to visualize the genotyping results and showed the polymorphism of these markers on gel.
Response: A representative gel picture has been added as suggested.
12. L296-L302: the marker descriptions on main text cannot match the supplementary table S8. Also, what are those 27 ILs on L298?
Response: Table S8 represents polymorphic markers between pairs of parents including some common polymorphic markers. Hence the total number represented in the table S8 is not additive and not matching with the numbers given in the text. 124 is the total polymorphic markers excluding repetition. 27 ILs were selected based on their agro-morphological similarity with recurrent parent ‘Krishna Hamsa’ and evaluated for background recovery. The list of 27 ILs has been included in the revised manuscript as suggested.
13. L602: there is no appendix.
Response: Mention of appendix at L602 is a typo error and has been removed